# Decoding Clonal Hematopoiesis: Emerging Themes and Novel Mechanistic Insights

**DOI:** 10.3390/cancers16152634

**Published:** 2024-07-24

**Authors:** Shalmali Pendse, Dirk Loeffler

**Affiliations:** 1Department of Hematology, St. Jude Children’s Research Hospital, Memphis, TN 38105, USA; 2Comprehensive Cancer Center, St. Jude Children’s Research Hospital, Memphis, TN 38105, USA; 3Department of Pathology & Laboratory Medicine, The University of Tennessee, Memphis, TN 37996, USA

**Keywords:** clonal hematopoiesis, CHIP, *Dnmt3a*, Tet2, Asxl1, inflammation, hematopoietic stem cells, self-renewal, clonal evolution, clonal selection

## Abstract

**Simple Summary:**

The expansion of mutant cells that outgrow and displace normal blood cells is called clonal hematopoiesis (CH). CH starts with the acquisition of mutations in blood stem cells that change their normal behavior and improve their fitness. Low numbers of mutant blood cells are present in many middle-aged and elderly people without noticeable effects. However, CH can progress to leukemia and contribute to diseases in other tissues, including the heart, liver, and pancreas. As these diseases progress and affect health when the number of mutant blood cells increases, preventing and/or reducing the expansion of mutant cells might delay this process. Identifying the factors driving mutant blood cell expansion is thus critical and has garnered heightened interest in the subject. Here, we review and discuss recent progress in the field that suggests that mutant blood stem cells are more resilient than normal cells and thrive in inflammatory conditions.

**Abstract:**

Clonal hematopoiesis (CH), the relative expansion of mutant clones, is derived from hematopoietic stem cells (HSCs) with acquired somatic or cytogenetic alterations that improve cellular fitness. Individuals with CH have a higher risk for hematological and non-hematological diseases, such as cardiovascular disease, and have an overall higher mortality rate. Originally thought to be restricted to a small fraction of elderly people, recent advances in single-cell sequencing and bioinformatics have revealed that CH with multiple expanded mutant clones is universal in the elderly population. Just a few years ago, phylogenetic reconstruction across the human lifespan and novel sensitive sequencing techniques showed that CH can start earlier in life, decades before it was thought possible. These studies also suggest that environmental factors acting through aberrant inflammation might be a common theme promoting clonal expansion and disease progression. However, numerous aspects of this phenomenon remain to be elucidated and the precise mechanisms, context-specific drivers, and pathways of clonal expansion remain to be established. Here, we review our current understanding of the cellular mechanisms driving CH and specifically focus on how pro-inflammatory factors affect normal and mutant HSC fates to promote clonal selection.

## 1. Introduction

The formation of blood relies on a small number of hematopoietic stem cells (HSCs) located in the bone marrow. HSCs can self-renew to create new stem cells and differentiate to produce all mature blood cell types for the entirety of the lifespan [1]. The ability of HSCs to regenerate the hematopoietic system throughout life is extensive, but it is not endless and the production of new blood cells changes and declines over time [2]. Contrary to that, mutant HSCs can self-renew indefinitely and are thought to drive hematological malignancies and therapy resistance [3]. These cells are sometimes called leukemic stem cells and their ability to self-renew is acquired by mutations in numerous genes affecting cell behavior and fates such as survival, proliferation, self-renewal, and differentiation. While these efforts are ongoing, and novel rare mutations are still being discovered, next-generation sequencing (NGS) studies with large cohorts of cancer patients identified the mutations that are most frequently associated with hematological diseases [4]. Surprisingly, these studies found that many of the somatic mutations associated with malignancies were already present in healthy individuals. Despite having no symptoms, healthy individuals had clonally expanded mutant hematopoietic cells [5,6]. As these mutations seemed to drive clonal hematopoiesis (CH) but did not lead to cytopenias and dysplastic hematopoiesis characteristic of other pre-malignant disorders, the benign clonal expansion was termed clonal hematopoiesis of indeterminate potential (CHIP) [5]. CHIP was originally defined as a subset of CH with a variant allele frequency ≥2% of somatic mutations in genes associated with hematological malignancies [7]. However, as we will discuss in more detail later, the 2% threshold is somewhat arbitrary and underestimates the frequency of CH, as smaller clones can be detected reliably. Importantly, although individuals with CH lack overt clinical symptoms, clonally restricted hematopoiesis increases the risk of myeloid [8] and lymphoid malignancies [9] as well as non-hematological diseases, including atherosclerotic cardiovascular disease, pulmonary disease, type 2 diabetes, acute kidney disease, chronic liver disease, and osteoporosis (Figure 1) [5,10,11,12]. 

Individuals with CH also have a higher risk for severe infection with COVID-19 and bacteria and show an increased overall mortality [13]. Due to its association with multiple hematological and non-hematological diseases, CH has gained considerable attention from the scientific community in recent years. Despite these efforts, many questions remain and the mechanisms regulating clonal expansion and progression from a benign preleukemic to a malignant leukemic state remain poorly understood. As the field moves forward quickly, we attempt to summarize what we have learned thus far, highlight key findings, and discuss how recent technological advances provide new insights and challenge old views. We also discuss the critical next steps and how new tools can help to improve our understanding of CH. 

## 2. Early Evidence for Clonal Expansion

In 1962, Mary Lyon proposed that one of the two mammalian X-chromosomes present in females is randomly inactivated during ontogeny [14]. This hypothesis, known today as X-chromosome inactivation (XCI), explained why female mice heterozygous for sex-linked genes have mosaic coat colors by suggesting that genetically identical cells (=clones) are present in patches with the same fur color. In the same year, Beutler et al. experimentally validated this theory in women heterozygous for the X-linked gene glucose-6-phosphate dehydrogenase (*G6PD*). Based on earlier observations, Beutler reasoned that if erythrocyte precursors of heterozygous women were a mixture of cells with normal and deficient G6PD enzyme activity, these mosaics should have intermediate G6PD activity in their blood. Using the stability of glutathione as a readout for G6PD activity, Beutler verified his hypothesis by comparing the glutathione stability of blood samples from heterozygous, hemizygous, and normal individuals with known family pedigrees for *G6PD* heterozygosity [15]. A few years later, in 1965, Linder and Gartler studied leiomyoma and myometrium tumor samples isolated from women heterozygous for two *G6PD* variants. Using starch gel electrophoresis, they found that the majority of tumors expressed the same G6PD variants, suggesting these tumors were derived from a single cell and of clonal origin [16]. Using the same system, Fialkow and colleagues studied blood samples of women who were heterozygous for *G6PD* with chronic myeloid leukemia (CML). Similar to Linder and Gartler, they found only one G6PD variant in the granulocytes and erythrocytes, while the skin fibroblasts expressed both G6PD forms, suggesting a clonal origin of CML [17]. In the 1990s, the development of novel assays, based on the high rate of polymorphism of the androgen-receptor gene made clonal studies based on XCI feasible in the majority of women by performing a simple PCR [18]. Using these assays on different tissues of the same women [19] and in women of different ages [20,21] revealed that clonal outgrowth (assessed by the strength of skewed XCI) is higher in the hematopoietic system and occurs more frequently in older individuals. In Champion et al., the XCI skewing of purified blood cells in normal elderly women showed that 63.4% of female patients with myeloproliferative disorder exhibit a clonal outgrowth in granulocytes and T cells. Based on these observations, he proposed clonal expansion might precede the development of myelodysplastic or myeloproliferative diseases [20].

## 3. Identifying the Driver of Clonal Expansion: The Genomic Revolution

While these early studies provided evidence for the expansion of hematopoietic clones, the analysis of G6PD variants and/or AR polymorphisms was limited, as they relied on low levels of heterozygosity in the general population and were of limited throughput, precluding the analysis of rare disorders and large cohorts [22]. Hence, the mutations and mechanisms driving CH remained largely unknown. This changed with the advent of affordable high-throughput sequencing, as mutations in blood cells of large cohorts of healthy individuals could be identified at scale with relatively little effort. Using these tools, in 2014, three independent groups discovered the most commonly mutated genes in benign and malignant hematopoietic clones [23,24,25], among which the loss-of-function mutations in the epigenetic modifiers *DNMT3A*, *TET2*, and *ASXL1* were most frequently observed (Figure 2A). Importantly, many CH mutations were shown to persist in AML patients after treatment and were found after relapse, supporting the idea that mutant clones precede malignancies [26,27].

Although some of the identified mutations in these studies were associated with lymphoid malignancies [24], their frequency was low. The majority of the field thus focused on the more abundant mutations that increase the risk for myeloid malignancies. However, in their recently published seminal work, Niroula et al. investigated 46,706 samples of healthy individuals and identified 234 mutated genes associated with an increased risk of chronic lymphocytic leukemia [28]. As the majority of these mutations did not give rise to myeloid malignancies, the authors classified the mutant variants into genes associated with myeloid clonal hematopoiesis (M-CH) and lymphoid clonal hematopoiesis (L-CH). Similar to M-CH, L-CH occurs more often in the elderly population. However, the overall frequency is lower and the most frequently found mutant variants in L-CH are more evenly distributed compared to M-CH [9] (Figure 2B). 

The size of these mutant clones is determined using the variant allele frequency (VAF), which represents the fraction of variant sequencing reads found in a genetic locus. In other words, VAF indicates how often a mutation is found in the sequenced DNA and provides an estimate of the relative number of mutant cells present in the analyzed sample. Based on a VAF cutoff of ≥2%, these early NGS studies identified a strong correlation between CH and age [23,24,29] and reported CH in 10–20% of individuals older than 70. At the same time, using the 2% threshold, benign clones were rarely found in younger individuals below 60 years of age at the time [30]. While a VAF of 2% rarely progresses to a malignant state, a growing body of evidence suggests that individuals with a VAF ≥10% (which corresponds to ≥20% of the nucleated blood cells with the mutation) exhibit a high risk of developing malignancies [31,32].

However, the sensitivity of these early sequencing studies was limited and clones with lower VAFs could not be detected reliably. More recent studies with improved sensitivity using ultra-sensitive sequencing [33], error-corrected sequencing [30], or single molecule molecular inversion probe sequencing [34], showed that *DNMT3A* and *TET2* mutant clones are detectable in 95% of studied cases in middle-aged individuals, suggesting that the acquisition of mutant clones is almost universal [30]. It is also increasingly becoming clear that mutant hematopoietic clones can be found decades before CH can be detected in the peripheral blood and that numerous independently expanded clones are present per individual [4]. 

Importantly, longitudinal studies following individual clones over 20 years showed that the originally proposed VAF > 2% threshold for CH can miss fast-growing, potentially harmful variants and that the number of mutant clones is highly variable over time and can grow, shrink, or remain steady (Figure 3) [34,35]. The phylogenetic reconstruction and analysis of mutant clones in monozygotic and dizygotic twins further showed that mutant clones can arise even in utero or during childhood. Interestingly, these studies did not find evidence of a genetic predisposition for CH [12], as suggested in earlier studies [36]. Another study that tracked individuals for a median time of 13 years and used phylogenetic reconstruction found that clones with different mutations expand at vastly different rates and at different times in life [12]. While the growth of mutant clones was overall exponential, clones with *DNMT3A* and *TP53* mutations were found to grow 5% per year, while clones with mutations in the splicing factor *SRSF2^P95H^* grew by over 50% annually. Interestingly, and for unclear reasons, clones with mutations in splicing factors appeared only later in life in aged individuals, while *TET2* clones emerged in young, middle-aged, and old age and expanded at a constant rate of 6.8% per year. *DNMT3A* mutant clones, on the other hand, grew faster early in life but progressively slowed down during aging [12]. 

These studies provide novel insights and question the still widely spread assumption that CH starts in elderly men and women. Instead, recent evidence suggests that, at least in some individuals, CH can start during development and/or early in life and that the presence of small mutant clones is universal in middle-aged individuals. These studies are also starting to reveal how different mutations affect clonal evolution and provide insights into the context-dependent role of environmental factors. These observations raise important questions, such as the following: Does CH always start during development? And if so, at what stage of blood formation do HSCs acquire these mutations? What is the cell of origin and why do these mutations occur in the first place? 

## 4. Cell of Origin

Because mutant clones persist for many decades and contribute to myeloid and lymphoid cells, it is thought that CH begins with mutations in HSCs [37]. However, the precise timing when these mutations occur remains poorly understood. While all blood cells are derived from a common ancestor preceding gastrulation [38], human HSCs form at the aorta–gonad–mesonephros region of the embryo at approximately 4–6 weeks after fertilization [39]. The HSCs then migrate to the placenta, fetal liver, and finally, the bone marrow, where they acquire the quiescent state typical for adult HSCs. While CH is still often thought to start with mutant HSCs in the adult bone marrow, recent phylogenetic analyses suggest that CH can start as early as in utero. Williams et al. estimated that the mutations in individuals with mutant *JAK^V617F^* and *DNMT3A* could have been acquired as early as 33 and 8 weeks of gestation, respectively, while a *PPM1D* mutation was acquired in an individual by 5.8 years of age [40]. Studies of mutant clones in monozygotic twins found rare somatic mutations can be shared in both [41], suggesting these mutations can occur as early as the morula or blastocyst stage, when the embryo consists only of a few dozen to a few hundred cells. These observations raise an important question: how representative are reports of clonal growth in the embryo and how often are mutations acquired early in life? As the phylogenetic reconstruction of genomic sequencing is a rather new approach [42] and relatively few individuals have been studied in great detail, the chances that extremely rare cases of embryonic clonal acquisition were discovered by chance seem unlikely. These findings thus suggest that the acquisition of mutations during embryogenesis might be far more common than expected. Indeed, estimates that cells acquire between 0.069 and 0.858 exonic mutations every year [43] and approximately 1–2 mutations per division during embryogenesis [38] suggest that the acquisition of mutations is the rule rather than the exception. Over time and many cell divisions, our bodies thus become complex genetic mosaics of clones with coding and non-coding mutations—a phenomenon known as somatic mosaicism—that was recently analyzed in the context of a *DNMT3A*-mutant individual with germline mosaicism [44]. For further reading on the causes and consequences of somatic mosaicism and its relevance to CH, we refer the reader to an excellent recently published review [45].

While somatic mosaicism and the acquisition of mutations in HSCs during development and adult life can create clones of varying fitness, another source of heterogeneity is HSC subtypes with different stable properties. While the exact number of HSC subtypes remains unknown, forms of myeloid-biased, lymphoid-biased, and HSCs with balanced lineage output have been reported by several laboratories in mice [46,47] and humans [48]. However, the relationship between HSC subtypes, somatic mosaicism, and clonal outgrowth is unclear, and whether mutations in different HSC subtypes affect clonal outgrowth, fitness, and risk for transformation has not been studied. As distinct HSC subpopulations react differently to TGF-β [49], context-specific effects might promote the clonal expansion of some mutant HSC subtypes but act inhibitory on others—a poorly understood phenomenon that requires further investigation. 

## 5. Mechanisms of Clonal Expansion

After a decade of research, and identifying the major mutations associated with CH, it is now well established that mutant HSCs possess a selective advantage over their normal counterparts [50,51,52]. Despite their fitness advantage, how CH is initiated and how mutant HSCs and their clones expand remain poorly understood [53]. Physiological stress, including repeated infections, autoimmune dysregulation, systemic disease, and metabolic stress promoting inflammation, have been suggested to provide the positive selection pressure required for the expansion of mutant clones [54]. Understanding these mechanisms and identifying the contributing molecular pathways is critical to developing therapeutic interventions that prevent clonal growth, disease progression, and malignant transformation.

As numerous mutations have been associated with CH, patterns of deregulated cellular pathways have emerged and mutant genes can broadly be categorized into regulators of (1) epigenetic modifications, (2) RNA splicing, (3) the DNA damage response, (4) transcription, (5) chromatin remodeling, and (6) signaling. In addition, CH genes can be categorized into (1) gain- and loss-of-function mutations and (2) M-CH and L-CH (Figure 2). The most commonly found mutations in *DNMT3A*, *TET2*, and *ASXL1* responsible for M-CH are epigenetic regulators and are loss-of-function mutations. *SF3B1*, *SRSF2*, and *U2AF1* are splicing factors [55], while *TP53* and *PPM1D* regulate the DNA damage response [56]. Besides these insights, the precise molecular mechanisms regulating clonal expansion, evolution, and disease progression remain ill defined [11]. L-CH and its associated mutations have been reviewed recently by von Beck et al. [9]. For the remainder of this review, we will thus focus on understanding M-CH-related mutations and specifically focus on our mechanistic understanding of *DNMT3A-*, *TET2-*, and *ASXL1*-mutant CH. 

Classically, HSC fates are thought to be regulated by cell-intrinsic and -extrinsic mechanisms. Intrinsic mechanisms are the inherent properties of the cell, such as the cell cycle, transcription factors, and epigenetic modification, while extrinsic mechanisms refer to signals from the microenvironment, including growth factors and cell–cell and cell–matrix interactions [57,58]. Although mutant HSCs rely on the same cell-intrinsic and -extrinsic processes as normal HSCs, some of these processes are deregulated, causing mutant HSCs to behave and respond differently to external cues. As these cell-intrinsic and -extrinsic processes are interrelated, it has been difficult to identify the key pathways and the precise order of events, as well as the cause and consequences of clonal expansion. It is thus maybe not surprising that clonal outgrowth appears to be highly context-dependent. Factors driving the expansion of one mutant clone might not affect or even inhibit the relative fitness of another [54]. Environmental factors including infection, inflammation, and chemotherapy have been shown to exert context-dependent effects and clonal outgrowth by increasing HSC fitness through changing survival, proliferation, self-renewal, or differentiation in response to different stressors—factors and mechanisms we will discuss in detail below. 

## 6. Inflammation and Clonal Hematopoiesis

A central role of the hematopoietic system is to fight infections by viruses, bacteria, and other insults that are detected by mature blood cells. When mature blood cells detect a threat, they secrete inflammatory cytokines to alarm neighboring cells to coordinate and amplify the immune response. This response typically involves changes in cellular activity, migration, cell divisions, and differentiation and also affects HSCs [59,60]. Normal HSCs, which are mostly quiescent and do not cycle, become activated, divide, and return to a quiescent state when the infection has passed and inflammatory cytokine levels drop [61]. Chronic inflammation, i.e., a situation when inflammatory cytokine levels do not return to baseline, impairs normal HSC function by inducing proliferative stress and premature exhaustion [62]. As inflammatory cytokine levels increase during aging [63], and CH correlates with age [64], identifying the drivers of inflammation and understanding how HSPCs and more mature blood cells respond to inflammatory cytokines have been of great interest and attracted a lot of attention in recent years. 

### 6.1. Mutant Mature Myeloid Cells Produce Aberrant Levels of Inflammatory Cytokines

Pro-inflammatory cytokines influence normal and mutant HSCs and are thought to promote the selective expansion of clones over time [53]. Increased levels of Interleukin-6 (IL-6), IL-1β, IL-12p70, IFN-γ, IL-4, IL-5, IL-10, TNF-α, and C-reactive protein, a marker for systemic inflammation, were found in the blood of individuals with CH [65,66]. As part of the immune system, terminally differentiated blood cells are a major source of inflammatory cytokines, suggesting that somatic mutations driving selective clonal expansion can also affect the behavior and function of mature cell types [67]. Indeed, macrophages, a class of myeloid cells that possess the ability to promote and restrain inflammation [68], secrete increased levels of inflammatory cytokines and promote the development of cardiovascular disease in *Tet2*-mutant cells in atherosclerosis-prone, low-density lipoprotein receptor-deficient (*Ldlr*^−/−^) mice. In line with this observation, *Tet2*-mutant macrophages are hypersensitive to low-density lipoprotein (LDL) or bacterial lipopolysaccharide (LPS) and IFN-γ and secrete more IL-1β and IL-6, a process mediated by the NLRP3 inflammasome [69]. The specific deletion of *Tet2* in myeloid cells, but not other lineages, in mice further suggests that the altered response of *Tet*^+/−^ or *Tet*^−/−^ macrophages to high levels of LDL (i.e., a high-cholesterol diet) is sufficient to increase aortic lesions and to induce atherosclerosis in *Ldlr* knock-out animals [32]. These insights thus provide a mechanistic link between *Tet2*-mutant CH and the increased risk of myocardial infarction [32]. Besides *Tet2*, other CH-associated mutations can also lead to elevated levels of pro-inflammatory cytokines. For instance, *Ppm1d*-mutant mouse macrophages produce more IL-1β and IL-18 after LPS stimulation in vitro [70], and macrophages in *asxl1-*, *tp53-*, and *dnmt8* (ortholog of DNMT3A)-mutant zebrafish produce more IL-1β and TNF-α [71]. While these findings suggest that mutant macrophages might be commonly deregulated in CH, evidence for increased inflammatory cytokine production was also found in *asxl1-* [71], *Tet2-* [72], and *Runx1*-mutant neutrophils [73], and *Dnmt3a*-deficient mast cells [74], which produce more IL-6, TNF-α, and IL-13 [74]. While only a few studies have looked into the role of mutant mast cells in CH, mutations in *DNMT3A*, *TET2*, or *ASXL1* were shown to affect the prognosis and reduce the overall survival of patients with mastocytosis, a hematological neoplasm with aberrant accumulation of mast cells in multiple organs [75,76]. Together, these studies suggest that elevated levels of pro-inflammatory cytokines found in individuals with CH are produced by the aberrant behavior and function of mature cell types, such as macrophages, neutrophils, and mast cells. Higher levels of pro-inflammatory cytokines produced by mutant mature cells are thus likely create a positive feedback loop to reinforce and/or stabilize an inflammatory environment that allows mutant hematopoietic cells to thrive.

Beyond simply secreting inflammatory cytokines, the cells of the innate immune system can also be in direct contact with HSCs. For instance, during trogocytosis, a quality assurance mechanism, macrophages ‘groom’ HSCs and remove cytoplasmic material from HSCs [77]. Macrophage ‘dooming’ leads to the engulfment and death of damaged HSCs with high levels of reactive oxygen species. The role of trogocytosis in CH is, however, unclear. 

### 6.2. Changes in the Adaptive Immune System

The aberrant behavior of mature blood cells with CH mutations is not restricted to myeloid cells as both the T and B cell function is altered by the loss of *Dnmt3a* or *Tet2* [78]. Studies of *Dnmt3a*-deficient T cells in mice, for instance, showed that DNMT3A is required to regulate cytokine production. In the absence of DNMT3A, T cells are unable to silence the expression of *Il2*, *Il4*, and *Infg* and produce elevated cytokine levels [79,80,81,82]. Conditional and B cell-specific deletion of *Dnmt3a* causes chronic lymphocytic leukemia [83] and the immunization of B cells deficient for *Dnmt3a* and *Dnmt3b* resulted in the antigen-specific expansion of B cells in germinal centers, plasma cell accumulation, and elevated levels of serum antibodies [84]. *Tet2* loss impairs the differentiation of B-2 cells into plasma cells [85] and increases B-1 cell numbers and IgM production in the bone marrow and spleen [86]. Besides these changes in differentiation, the disruption of either *Dnmt3a* [87] or *Tet2* [88] was found to promote the therapeutic efficacy of CAR T cells, suggesting that these mutant CAR T cells do not become exhausted and maintain their capacity to proliferate. This prolonged anti-tumor activity might be leveraged to improve future CAR T cell therapies.

### 6.3. An Anti-Inflammatory Program Protects Mutant HSPCs

Contrary to mutant mature cells, which upregulate pro-inflammatory programs, recent observations suggest that *DNMT3A-*, *TET2-*, and *ASXL1*-mutant HSCs upregulate anti-inflammatory programs to counteract the effects of inflammation. Although systematic studies on the effects of pro-inflammatory cytokines are lacking, and some of the reported effects of inflammation on mutant HSCs are not consistent, overall, the following model emerges. Inflammation depletes and/or impairs the self-renewal capacity of normal HSCs [62,89], while mutant HSCs evade and/or suppress inflammation to improve their fitness in conditions detrimental to normal HSCs (Figure 4). 

In a zebrafish model of *asxl1* CH, the fitness advantage of mutant HSPCs depends on the upregulation of the negative regulators of inflammation *socs3a* and *nr4a1* [71], as the homozygous deletion of *nr4a1* alone was able to prevent clonal outgrowth (Figure 5A). This demonstrates that *nr4a1* is required for the competitive advantage of *asxl1*-mutant HSCs [71]. The upregulation of *Socs3* and *Nr4a1* was also found in *Dnmt3a^fl-R878H/+^* [90] and in *VavCre;Tet2^fl/fl^* [91] mutant HSCs in mice, suggesting that the outgrowth of different mutant clones might share mechanisms to improve their fitness, possibly by increasing their resilience to inflammatory stress (Figure 6C and Figure 7C). However, in a mouse model of chronic infection using *M. avium*, IFN-γ signaling led to the downregulation of *Socs3* and *Nr4a1* in *Dnmt3a*^−/−^ HSPCs [92]. At this point, it thus remains unclear whether these discrepancies are caused by cytokine-mediated context-specific effects or if HSCs adapt and execute different programs to boost resilience in response to chronic inflammation mediated by a ‘real’ infection (Figure 6D). While these discrepancies and presumably context-specific effects require more studies, it is well established that normal HSCs need to suppress inflammation and proliferative stress to prevent exhaustion. The ability to suppress inflammation might be limited in normal HSCs while mutant HSCs can suppress inflammation more efficiently and/or for longer. In line with this hypothesis, the loss of negative regulators of inflammation, such as the Activating Transcription Factor 3 (ATF3), leads to HSC exhaustion after 5-fluorouracil (5-FU) treatment of *VavCre;Atf3^fl/fl^* mice [93]. The nuclear orphan receptor NR4A1 restricts HSC proliferation and DNA damage and represses inflammation via C/EBPα [94], while the loss of SOCS3, a negative regulator of cytokine signaling, increases inflammation [94]. Importantly, *NR4A2* and other anti-inflammatory genes are also upregulated in HSCs isolated from individuals with *DNMT3A* and *TET2* CH [95]. It thus seems plausible that upregulation of *ATF3*, *SOCS3*, *NR4A1*, and other anti-inflammatory genes could be beneficial and protect mutant HSCs. If correct, how precisely do these anti-inflammatory programs protect mutant HSCs? Recent studies, mostly using genetic models of CH in mice, started to shed light on this question and suggest that complex context-dependent but interrelated mechanisms affect HSC survival, proliferation, differentiation, and self-renewal to improve fitness in *Asxl1* (Figure 5), *Dnmt3a* (Figure 6), and *Tet2* mutants (Figure 7) [45]. However, how individual inflammatory cytokines affect mutant HSC fates remains poorly understood.

## 7. Changes in Normal and Mutant HSC Fate Decisions

While inflammation is advantageous to mutant HSCs (Figure 4), the precise role of HSC fate decisions contributing to clonal expansion remains poorly understood. Given that mutant HSCs are typically present for decades without noticeable outgrowth and/or consequences, the initial changes in HSCs are likely subtle. However, even slight differences in mutant vs. normal HSCs’ survival in response to stress can amplify over many decades and cause large differences in the relative size of clones. Although this process starts slowly, it progressively accelerates over time, very much like earning interest on savings in a bank account. As the yearly interest earned is based on the cumulative earnings of prior years, the largest ‘gains’ are typically obtained many years later in life. Although this model is a helpful analogy to understand how small changes in HSC numbers can lead to large clonal outgrowth many years later, it is important to remember that clonal competition is more complex. The rates of clonal expansion (i.e., the ‘interest rate’) can change over time as the cells adapt to new stresses and can also involve changes in HSC proliferation, self-renewal, and differentiation. In line with these considerations, recent evidence from longitudinal studies shows that clone numbers can shrink and/or remain stable in size for years (Figure 3). But how do mutant clones change these fate decisions to increase their fitness, and can we use these insights to create new therapies?

### 7.1. Improved Survival of Mutant Clones

The most intuitive way to improve cellular fitness might be increased cell survival. A reduction in cell death can, either under steady-state conditions or stress, lead to a shift in the relative frequencies of clones as the surviving cells can divide again to expand further. Increased cell survival as a mechanism of clonal expansion has been described for *PPM1D*-mutant clones [104]. In this study, Hsu et al. treated *PPM1D*-mutant clones in vitro and in vivo with cytotoxic agents and discovered that among others, cisplatin treatment led to a relative expansion of *PPM1D*-mutant cells. As PPM1D suppresses P53, it is not surprising that *PPM1D*-mutant clones partially phenocopy *p53*-mutant HSCs, which outcompete their normal counterparts [105]. Whether changes in survival affect mutant HSCs with other mutations that are not directly linked to *p53* is, however, less clear. In a tamoxifen-inducible *Tet2*^+/−^ and *VavCre;Tet2*^fl/fl^ model of *Tet2* CH, cell death between *Tet2*-mutant and wild-type HSCs, before and after treatment with IL-1α [102] and IL-1β [91], respectively, was comparable (Figure 7C,D). However, other studies reported lineage-negative *Tet2*^−/−^ cells died less after TNF-α [101] and IL-6 stimulation [100] (Figure 7A,B). In the latter study, increased levels of IL-6 induced hyperactive SHP2-STAT3 signaling, which resulted in the upregulation of the anti-apoptotic long non-coding RNA *morrbid* in *Tet2*-KO myeloid cells and progenitors. In general, these studies suggest that the improved survival of *Tet2*-mutant HSPCs in response to IL-6 and/or TNF-α is mediated by the increased expression of anti-apoptotic *Bcl2* and *Birc2*, while the pro-apoptotic genes including *Bcl2l11*, *Casp1*, *Bcl2l1*, *Fas*, *Casp3*, and *Casp8* are downregulated [100]. However, as both studies used *Tet2*-mutant myeloid cells and their progenitors, whether this mechanism also extends to highly purified HSCs remains unclear. Similar to *Tet2*, also, *Dnmt3a^−/−^*- and *Dnmt3a^R878/+^*-mutant HSCs show increased survival in response to inflammatory cytokines. However, contrary to *Tet2* mutants, *Dnmt3a*-mutant HSPC survival is unaltered and comparable to that of normal HSPCs exposed to IL-6, TNF-α, and IFN-α [99]. However, *Dnmt3a*-mutant HSPCs were more resilient to IFN-γ-mediated stress as suggested by reduced Caspase 3/7 luminescence levels in vitro (Figure 6D,E). Although the survival of *Dnmt3a*-mutant HSCs in vivo was unaltered, similar to *Tet2*, *Dnmt3a*-mutant HSPCs had elevated levels of *Bcl2*. However, as the *Bcl2* levels were already increased in *Dnmt3a*-mutant cells before inflammatory cytokine stimulation, it is surprising that *Bcl2* did not protect against the deleterious effects of other inflammatory cytokines. The interpretation of this these data is not straightforward, and further, more systematic studies using more sensitive tools with single-cell resolution are needed to clarify the role of cell death in response to individual cytokines in mutant HSPC survival.

### 7.2. Changes in Proliferation of Mutant Clones

Besides changes in survival, mutant HSC clones can also expand by increasing proliferation. This can either be accomplished by shortening the overall cell cycle duration between divisions and/or by increasing the number of divisions before HSCs return to quiescence after entering the cell cycle. Overall, few studies have looked into how the proliferation and/or cell cycle kinetics of highly purified mutant HSCs change [71], and the context-specific changes in response to different cytokines are poorly understood. In line with previous studies, using a *Tet2*^+/−^ mouse model, Caiado et al. observed the expansion of myeloid-biased multipotent progenitors with a simultaneous reduction in the numbers of downstream Lin^−^cKIT^+^ progenitors 14 days post IL-1α injection (Figure 7D). Although fewer *Tet*^+/−^ HSPCs remained in G_0_ after the IL-1α injections, the overall frequencies of the HSPCs did not change compared to the wild-type animals treated with IL-1α [102]. *Tet2*^−/−^ HSPCs isolated from *VavCre;Tet2*^fl/fl^ mice after IL-1β injection show similar trends, with no to minor changes in HSPCs in G_0_ and a slight increase in HSPCs in S/G_2_, suggesting *Tet2*-mutant HSCs proliferate more [91]. While these studies suggest Tet2-mutant HSPCs proliferate slightly more than their wild-type counterparts, how these seemingly small changes in proliferation account for the observed expansion of immunophenotypic HSCs remains obscure. While increased proliferation might be expected, it is not required for clonal outgrowth. As shown recently by the Challen group, where at least in the context of IFN-γ, highly purified *Dnmt3a*-mutant HSCs seem to cycle less. Although key insights are still missing, it appears that inflammatory cytokines tend to induce the proliferation of *Asxl1-* (Figure 5) and *Tet2*-mutant HSPCs (Figure 7), while the proliferation of *Dnmt3a* mutants remains unaltered or slows down (Figure 6C–E).

### 7.3. Self-Renewal and Differentiation

Although a reduction in the proliferation of *Dnmt3a*-mutant HSCs might seem counterintuitive to accomplish clonal expansion, excessive proliferation is not required when self-renewal increases or differentiation decreases (Figure 8). At first glance, increased self-renewal and decreased differentiation seem to be the same as both lead to clonal expansion. However, the identification of self-renewal and differentiation gene signatures suggests that there are different programs and that HSCs can expand by either increasing self-renewal and/or by reducing differentiation [106]. Evidence from multiple independent groups has firmly established that both *Dnmt3a-* [107] and *Tet2* [108]-mutant mouse HSCs have increased self-renewal and outcompete wild-type HSCs in competitive transplants in vivo. The self-renewal capacity of *Dnmt3a*-mutant HSCs thereby seems to outcompete HSCs with mutations in *Tet2* [109]. The Challen lab illustrated this by showing that *Dnmt3a*KO HSCs can be serially transplanted at least 12 times without exhaustion [110]. *Tet2*-deficient HSCs, on the other hand, have been suggested to lose their self-renewal ability and become exhausted after tertiary transplant [109]. Although a detailed mechanistic understanding of how these mutations lead to increased self-renewal and/or differentiation is lacking, a molecular analysis of *Dnmt3a-* and *Tet2*-mutant HSCs showed progressive and localized reductions in DNA methylation at crucial regulatory sites linked with self-renewal genes [110]. Compared to *Dnmt3a* and *Tet2*, how *Asxl1* mutations affect self-renewal and differentiation is less clear. Similar to individuals with CH, *Asxl1*-mutant mouse HSCs expand with age [96]. However, contrary to *Dnmt3a* and *Tet2* mutants, *Asxl1*-mutant HSCs do not outcompete wild-type HSCs in competitive transplants [96]. This suggests that transplant data, which are often referred to as the ‘gold standard’ to assess HSC function, should be interpreted carefully, as transplantation might expose mutants and/or wild-type HSCs to stress they would not experience under normal circumstances.

Taken together, understanding the mechanisms regulating the survival, proliferation, self-renewal, and differentiation of normal and mutant HSPCs is critical to developing novel therapies, as altering HSPC decisions can slow down and/or prevent clonal expansion and disease progression. As mutant HSPCs seem to be more resilient to inflammatory cytokines than normal HSPCs, reducing inflammation therapeutically should reduce the risk of developing hematological and non-hematological diseases associated with CH (Figure 1). If successful, these therapies might lower the risk of many diseases, including cardiovascular diseases, one of the leading causes of death.

## 8. DNA Methylation

The DNA methylation of cytosine–guanine dinucleotides, also referred to as “CpG sites”, is crucial for regulating gene expression, chromatin accessibility, and nuclear architecture [112] and is mediated by DNMT1, DNMT3A, and DNMT3B [113]. Changes in DNA methylation are a hallmark of cancer [114] and were found in HSCs harboring CH mutations, including mouse models of *Dnmt3a*^−/−^ [107], *Dnmt3a^R878H/+^* [115], *Tet2^−/−^* [116], *Tet2^+/−^* [117], and *Asxl1* [118]. These studies showed that DNA methylation is largely maintained throughout the genome in mutant HSCs. However, specific genomic regions are differentially methylated. Based on their overall size and ‘shape’ in the genomic landscape, these regions have been termed CpG islands [119], shores [120], shelves [121], canyons [122], and the more recently described mesas [123]. CpG islands are smaller regions approximately 1000 bp long with little or no DNA methylation and are typically located at promoters, gene bodies, and enhancer regions and make up approximately 1–2% of the mammalian genome [124,125]. CpG shores span less than 2 kb and flank CpG Islands, and CpG shelves span less than 2 kb and flank outwards from CpG shores [122,126]. Approximately 1000 DNA methylation canyons in the genome represent regions with low average methylation levels that are larger than 3.5kb [113], while mesas or promotor exon 1—intron 1 (PrExI) regions are suggested to act as demethylation firing centers [123].

Paradoxically, *DNMT3A* and *TET2*, the first and second most frequently mutated genes in CH, respectively, have opposing roles but lead both to increased HSC self-renewal and promoted clonal outgrowth. DNMT3A adds methyl groups, and TET2 demethylates CpGs by adding a hydroxyl group and converting methylcytosine into 5-hydroxymethylcytosine [127]. The detailed mechanism of how site-specific changes in DNA methylation regulate self-renewal remains to be elucidated, but several observations provide clues. For instance, in *Dnmt3a*KO (*Dnmt3a^−/−^*) mouse HSCs, DNA methylation at enhancers and canyons is reduced [128]. As these canyons are enriched for genes associated with HSC self-renewal, including *HoxA9*, *Meis1*, and *Evi1*, reduced DNA methylation and increased expression of self-renewal genes in *Dnmt3a^−/−^* HSCs could be linked to changes in HSC function. Contrary to WT HSCs, which cease to engraft after 3–4 rounds of serial transplantation, *Dnmt3a*^−/−^ HSCs can be transplanted for at least 12 consecutive rounds [110]. Based on these observations, Challen and Goodell [52] proposed that when instructed to differentiate, DNMT3A silences the HSC self-renewal program to allow differentiation. In the absence of DNMT3A or in CH, where DNMT3A function is greatly reduced, HSCs cannot efficiently differentiate. While this model is appealing, many questions remain. How DNMT3A localizes and targets promoters and enhancers associated with HSC-specific genes for DNA methylation and silencing is poorly understood. It also appears that one would expect a more severe phenotype if differentiation was less efficient than proposed. However, even complete *Dnmt3a*KO HSCs are still able to differentiate and produce mature blood cells. Also, if the lack of DNMT3A increases self-renewal because self-renewal genes cannot be shut down, is the increased fitness of *Tet2*-mutant HSCs mediated because genes required for effective differentiation cannot be turned on? While the answer to this question is unclear, changes in DNA methylation might provide additional insights [129,130]. Studies using algorithms to analyze changes in the DNA methylation of blood cells during aging have shown that the biological age of individuals can be predicted with high accuracy [131,132]. These algorithms, including Horvath [133], Hannum [134], PhenoAge [135], and others, have been termed DNA methylation clocks and have recently been used on blood samples from individuals with CH and hematological malignancies [136,137]. A study of 154 Danish twins using the Hannum and GrimAge clocks found that the biological age of individuals with CH is accelerated when compared to individuals without mutant clones [138]. Accelerated aging was also found in individuals with mutations in *DNMT3A* and *TET2* by Nachun et al., who studied 5522 individuals [136]. While clinical studies using DNA methylation clocks to assess the biological age of individuals relied on the analysis of bulk blood cells isolated from the peripheral blood, Mick Milsom’s group recently showed for the first time that DNA methylation clocks are also accelerated in highly purified HSCs in a mouse model of chronic inflammation [62]. Similar studies of mutant HSCs might help to decipher how changes in DNA methylation regulate genes required for HSC self-renewal and differentiation. 

## 9. Conclusions

Ten years after the discovery of genetic variants in the peripheral blood cells of healthy elderly individuals, the phenomenon of clonal hematopoiesis, in essence, the expansion of mutant HSC clones, has taken the stem cell field by storm [23,24,29]. The heightened interest is fueled by the opportunity and desire to discover and understand the very origins of hematological diseases, and the hope that these efforts will lead to the development of novel early therapeutic interventions. These research efforts promise nothing less than the possibility to detect and fight potential mutant HSC clones before they progress to a more difficult-to-treat malignant state. After the discovery that CH affects both hematological and non-hematological diseases, including cardiovascular disease and diabetes [139], which, together, claim millions of lives worldwide every year [140], the interest in CH exploded as the eradication of mutant blood clones might help to treat and/or prevent a plethora of diseases (Figure 1). Although we have seen tremendous progress in the last decade, we are still in the early stages of understanding the ramifications of this phenomenon and many mechanisms remain to be discovered.

Despite many open questions, inflammation is emerging as a common theme and driver of clonal expansion. Recent evidence suggests that mutant HSCs are more resistant to the effects of inflammation than normal HSCs [71,92] and that mutant mature cells, including macrophages, neutrophils, mast cells, and others, fuel disease progression by producing more inflammatory cytokines than normal mature cells [141]. Elevated levels of the classical pro-inflammatory factors of IFN-γ, IL-1β, IL-6, and TNF-α have been repeatedly reported in both mouse models [91] and individuals with CH [142], but additional less well-studied factors such as Oncostatin M are involved, too [90]. However, systematic studies that map how mutant HSCs and their offspring differ from normal cells in their response to cytokines are lacking. Why mutant myeloid cells produce more cytokines is incompletely understood, but a cell-intrinsic pro-inflammatory profile and a hyper-responsive phenotype to environmental cues have been proposed [53]. These studies highlight that external factors, including infections [143], smoking [144], and prior exposure to chemotherapy [104], promote CH. The biological consequences of these cues are likely highly context-dependent and will vary depending on the specific mutation and genetic predispositions, as recently described for an inherited polymorphism in the *Tcl1a* promotor [145]. Despite these insights, whether CH is induced by an existing pro-inflammatory state or creates the pro-inflammatory environment required for clonal expansion remains unclear. As inflammatory cytokines such as IFN-α were suggested to induce asymmetric cell division (ACD) in normal HSCs [146], changes in cell polarity and ACD might play a role in regulating the self-renewal and differentiation of mutant HSCs. ACD is an important and evolutionarily conserved process that regulates metabolic activation and the return to quiescence after HSC activation through the asymmetric mitotic segregation of lysosomes and other organelles [147,148,149,150]. How inflammation affects the cell polarity and division of mutant HSCs remains poorly understood and/or has not been studied, but might provide important clues to understand how mutant HSCs expand. So far, sequencing technologies have been at the forefront of discovery and novel and more sensitive approaches such as error-corrected targeted sequencing [151], deep-targeted sequencing [152], and ultra-sensitive sequencing [34] are now available to detect even small clones with a VAF below 0.01%. Also, computational tools developed to analyze clonal dynamics and the sequential acquisition of mutations over the human lifespan continue to provide new insights [78,79]. While sequencing will continue to shape our understanding of CH and novel rare mutations will be identified in the years to come, the discovery of new mechanistic insights as a basis for novel targeted therapies will require that the field moves beyond ‘simple’ sequencing studies. In the not-so-distant future, hundreds of millions of genomes will have been sequenced. Sequencing a few more genomes will then seem insignificant as we quickly approach a world where sequencing blood samples will be part of annual routine health checks. Emerging single-cell technologies, such as multi-omics approaches that integrate gene expression, epigenetics, and mutations, will continue to shape our understanding of clonal evolution [153]. However, lineage tracing tools to reconstruct clonal dynamics over human lifespans [4,111], and the direct observation of clonal evolution over many weeks and months using novel long-term live cell imaging [148,149], will be crucial moving forward and will likely be the new frontier.

## Figures and Tables

**Figure 1 cancers-16-02634-f001:**
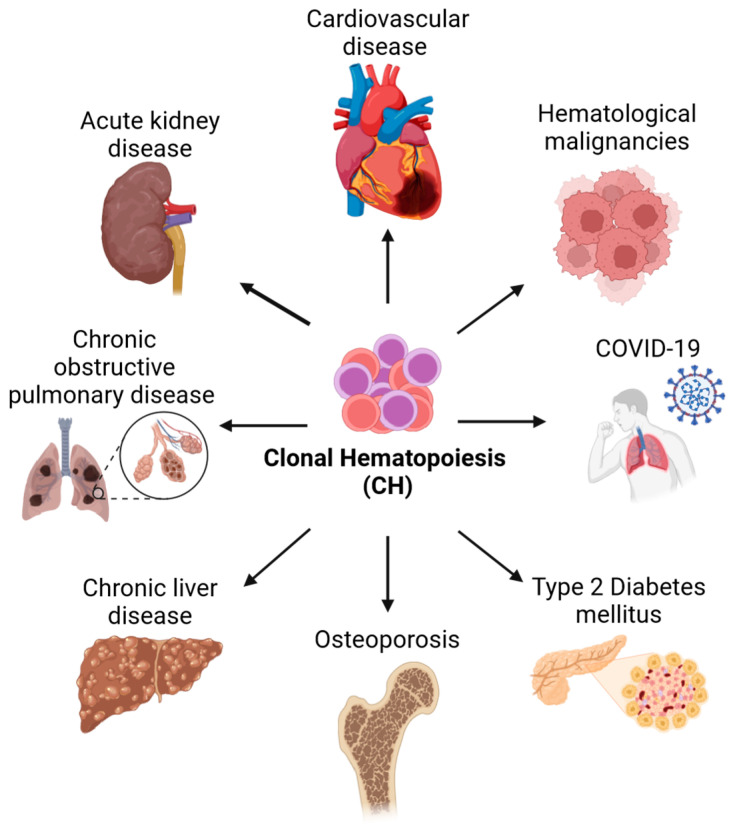
Diseases associated with clonal hematopoiesis. Individuals with clonal hematopoiesis have a higher risk of hematologic and non-hematologic diseases and an overall higher rate of mortality. Hematologic diseases associated with CH include myelodysplastic syndrome and myeloid and lymphoid malignancies. Non-hematologic diseases associated with CH include, among others, cardiovascular disease, acute kidney disease, chronic obstructive pulmonary disease, chronic liver disease, osteoporosis, type 2 diabetes, and COVID-19. Created with https://biorender.com.

**Figure 2 cancers-16-02634-f002:**
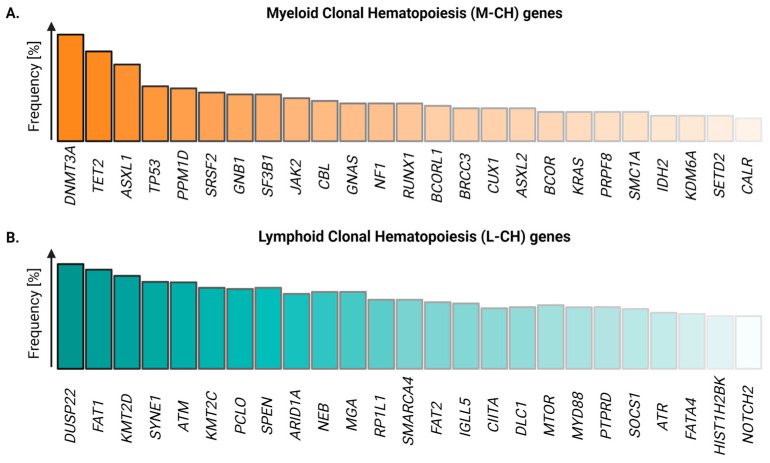
Frequencies of the most common somatic variants associated with myeloid and lymphoid clonal hematopoiesis. (**A**) The top 25 driver genes mutated in myeloid CH (M-CH) and (**B**) lymphoid CH (L-CH), respectively (adapted from [28]). Created with https://biorender.com.

**Figure 3 cancers-16-02634-f003:**
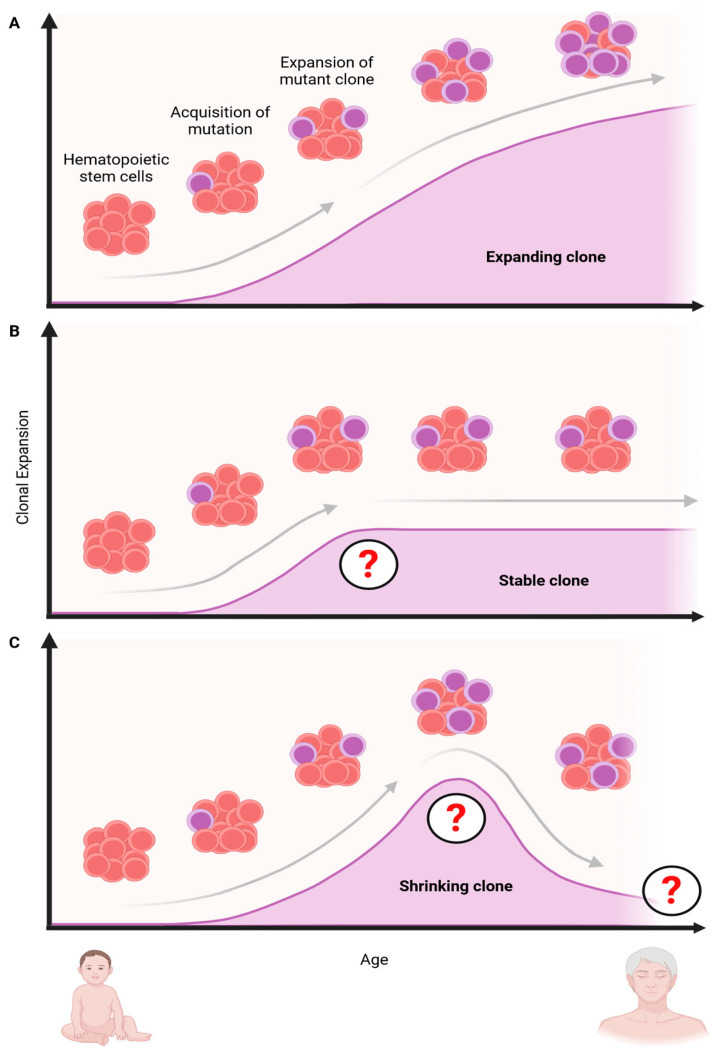
The number of mutant clones throughout an individual’s lifetime can change dynamically and grow, shrink, or remain stable for extended periods. Clonal hematopoiesis is initiated when HSCs acquire mutations that confer a competitive advantage over wild-type HSCs. (**A**) Over time, this mutation allows the mutant clone (blue) to expand relative to normal HSCs (red). (**B**) Alternatively, mutant clones can remain stable for extended periods and/or the entire lifespan. (**C**) Mutant clones that started to expand can also shrink again. This can happen if the environmental conditions become less favorable and/or if another clone with increased evolutionary fitness outcompetes the original clone. The red question mark indicates changes in clonal growth rates. The reasons for these changes are currently unclear. Created with https://biorender.com.

**Figure 4 cancers-16-02634-f004:**
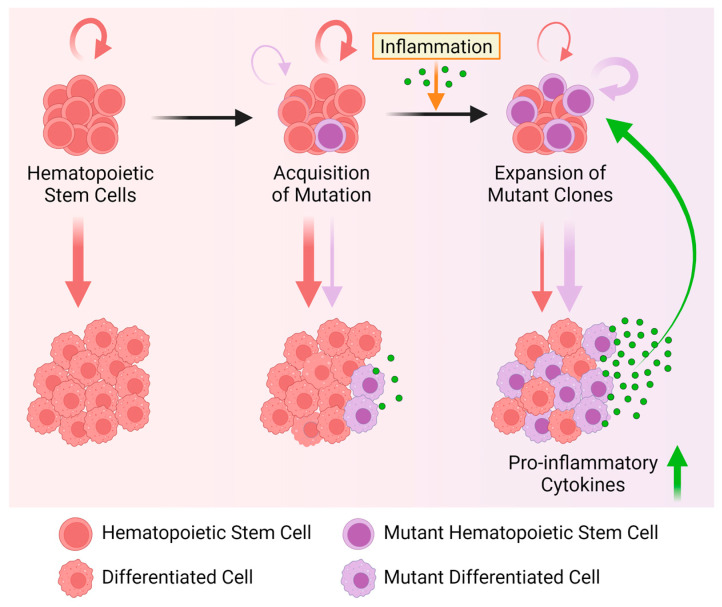
Inflammation drives clonal expansion of mutant HSCs. Normal HSCs (round red cells) self-renew to maintain the stem cell pool and differentiate to produce mature blood cells (left). Eventually, HSCs acquire mutations. While most mutations remain inconsequential, others provide a fitness advantage (round blue cells) under certain circumstances. In the absence of stress, these mutant HSCs self-renew and differentiate similarly to normal HSCs (middle). When exposed to inflammatory factors, mutant HSCs can outcompete normal HSCs through different cytokine-mediated and context-dependent mechanisms. The following clonal outgrowth of mature mutant blood cells leads to aberrant production of pro-inflammatory cytokines, reinforcing inflammation. As mutant HSCs are more stress-resilient in this pro-inflammatory environment, this selective advantage further promotes clonal expansion and perpetuates the cycle. Created with https://biorender.com.

**Figure 5 cancers-16-02634-f005:**
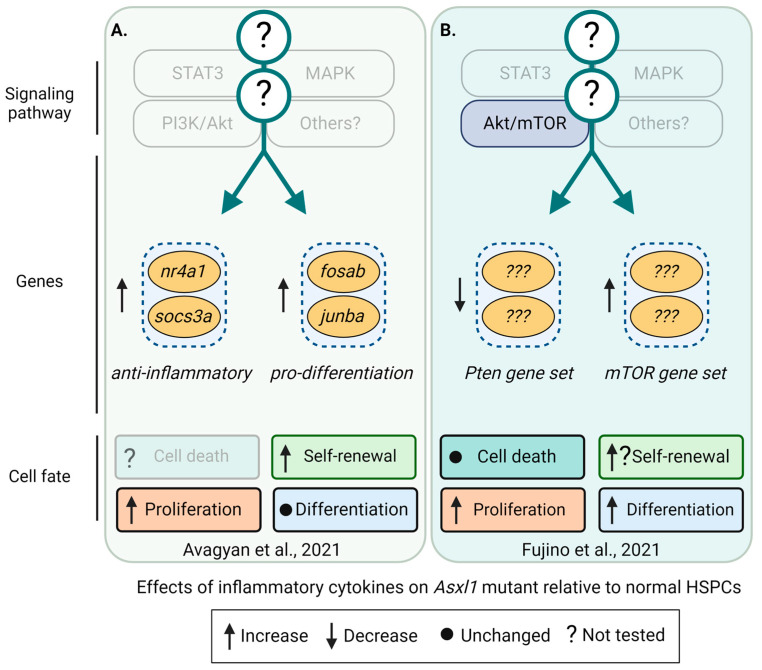
Differentially regulated pathways, genes, and cell fates in *Asxl1*-mutant HSPCs. Differentially regulated signaling pathways, genes, and their effects on cell fates, including cell death, self-renewal, proliferation, and differentiation of *Asxl1* mutant relative to normal HSPCs. Cell-extrinsic factors driving the clonal expansion of *Asxl1* mutants are currently unknown. Transparent boxes and annotations indicate unknown signaling pathways, genes, and cell fates. (**A**) In a zebrafish model, *asxl1*-mutant HSPCs exhibit increased proliferation and self-renewal. These cells express higher levels of anti-inflammatory *nr4a1* and *socs3a* and the pro-differentiation genes *junba* and *fosab.* Deletion of *nr4a1* showed that the fitness advantage of *asxl1*-mutant clones depends on *nr4a1* [71]. (**B**) Using a mutant *Asxl1* knock-in mouse model, clonal outgrowth and increased proliferation of *Asxl1*-mutant HSPCs were mediated by Akt/mTOR activation. However, transplants impaired *Asxl1*-mutant HSPC function [96]. Created with https://biorender.com.

**Figure 6 cancers-16-02634-f006:**
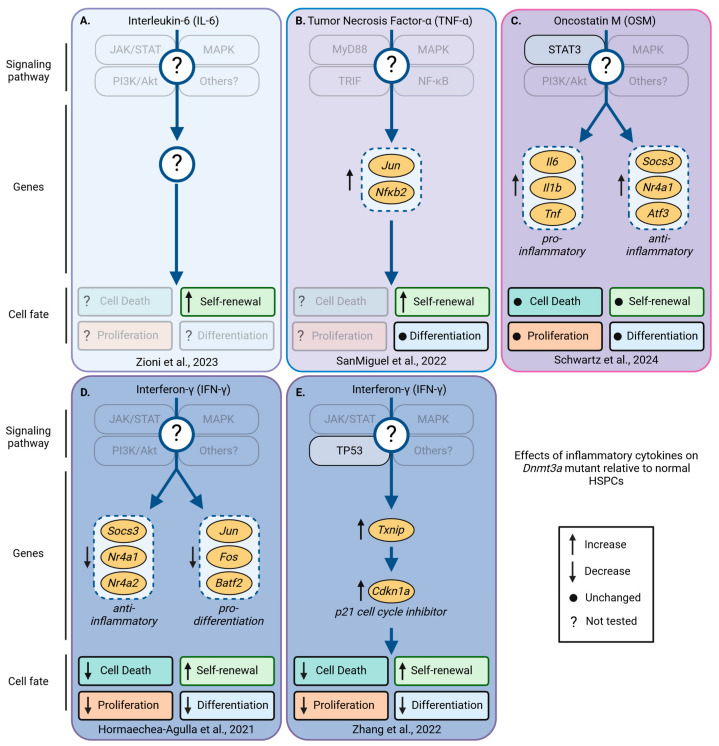
Pro-inflammatory cytokine-induced changes in signaling, gene expression, and cell fates that occur in *Dnmt3a*-mutant, but not in wild-type HSPCs. *Dnmt3a*-mutant and normal HSPCs respond differently to inflammatory cytokines. Depicted are the currently known differentially regulated signaling pathways, genes, and their effects on survival, proliferation, self-renewal, and differentiation of *Dnmt3a*-mutant relative to normal HSPCs in response to inflammatory cytokines. Several genes, including *Socs3*, *Nr4a1*, *Fos*, *Jun*, and *JunB*, are deregulated in *Dnmt3a*-mutant HSPCs; these genes are sometimes up- or downregulated, suggesting context-dependent effects of pro-inflammatory cytokines. Transparent boxes and labels indicate unknown pathways, genes, and/or HSPC fates not investigated. (**A**) In fatty bone marrow, elevated levels of IL-6 promote *Dnmt3a^fl-R882H/+^* HSPC self-renewal [97]. (**B**) Knock-out of TNF-α receptor 1 (TNF-αR1), but not TNF-αR2 increases HSPC self-renewal, without changing myeloid vs. lymphoid differentiation. TNF-αR1 signaling induces *Jun*, *Nfkb2*, and *CD69* more strongly in *Dnmt3a^R878H/+^* than in normal HSPCs [98]. (**C**) In young *Dnmt3a^R878H/+^* HSPCs, OSM activates STAT3, induces the expression of pro- and anti-inflammatory genes, and inhibits *JunB* expression. Despite these changes, OSM did not change HSPC fates, including cell death, proliferation, self-renewal, and differentiation [90]. (**D**,**E**) In response to IFN-γ, *Dnmt3a^−/−^* HSPCs die less and proliferate slower than wild-type HSPCs. In addition, *Dnmt3a^−/−^* HSPC self-renewal increased and differentiation decreased compared to wild-type HSPCs exposed to IFN-γ. (**D**) *M. avium* infection-mediated IFN-γ induces pro-inflammatory genes and inhibits pro-differentiation genes at the same time [92]. (**E**) IFN-γ induces TP53 signaling in *Dnmt3a^−/−^* HSPCs and inhibits cell cycle progression via Txnip-p21 [99]. Created with https://biorender.com.

**Figure 7 cancers-16-02634-f007:**
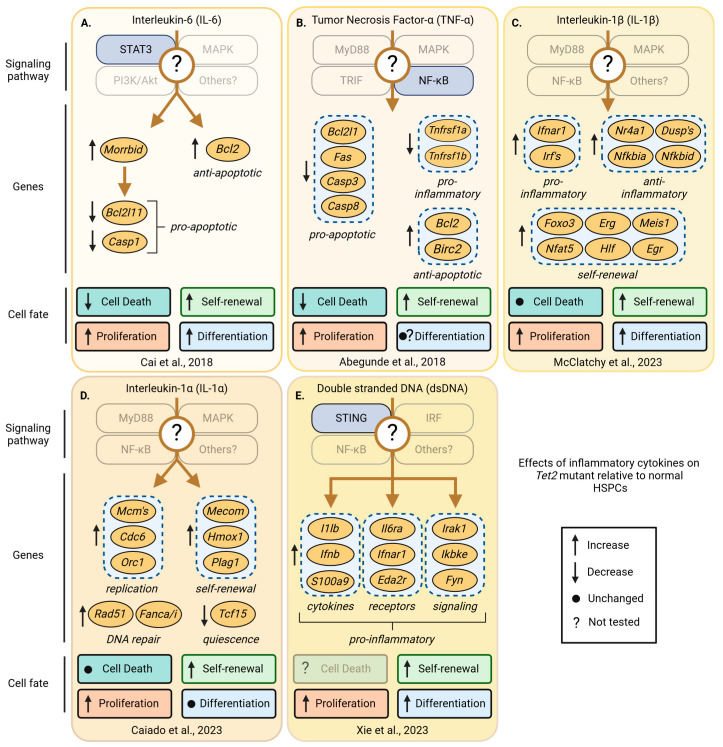
Pro-inflammatory cytokine-induced changes in signaling, gene expression, and cell fates that occur in *Tet2*-mutant, but not in wild-type HSPCs. The effects of inflammatory cytokines on *Tet2*-mutant HSPCs across multiple studies show that changes in signaling, gene expression, and HSPC fates vary between cytokines and relative to normal HSPCs. Shown are known differentially regulated signaling pathways, changes in gene expression, and their effects on *Tet2*-mutant relative to wild-type HSPC fates including cell death, self-renewal, proliferation, and differentiation after exposure to inflammatory cytokines. *Tet2*-mutant HSPCs have increased self-renewal and proliferate more compared to wild-type HSPCs in response to inflammatory signals. Transparent boxes and labels indicate unknown effects on signaling pathways, genes, and HSPC fates. (**A**) IL-6 mediated downregulation of pro- and upregulation of anti-apoptotic genes reduces cell death of *Tet^+/−^* and *Tet2^−/−^* HSPCs [100]. (**B**) TNF-α-induced NF-κB signaling reduces cell death in *Tet2^−/−^* relative to wild-type HSPCs and correlates with the up-and downregulation of anti- and pro-apoptotic genes, respectively [101]. (**C**) IL-1β induces the expression of pro- as well as anti-inflammatory genes in *Vav-Cre Tet2^fl/fl^* HSPCs [91]. (**D**) The increase in self-renewal, proliferation, DNA replication, and DNA repair gene expression programs of *Tet2^+/−^* HSPCs depends on IL-1α receptor 1-mediated signaling [102]. (**E**) Studies using *Tet2^fl/fl^;Sting^−/−^* mice show that the cGAS-STING pathway is required for the competitive advantage of *Tet2^−/−^* HSPCs. Genes encoding pro-inflammatory cytokines, receptors, and signaling factors depend on STING activation in *Tet2^fl/fl^* HSPCs [103]. Created with https://biorender.com.

**Figure 8 cancers-16-02634-f008:**
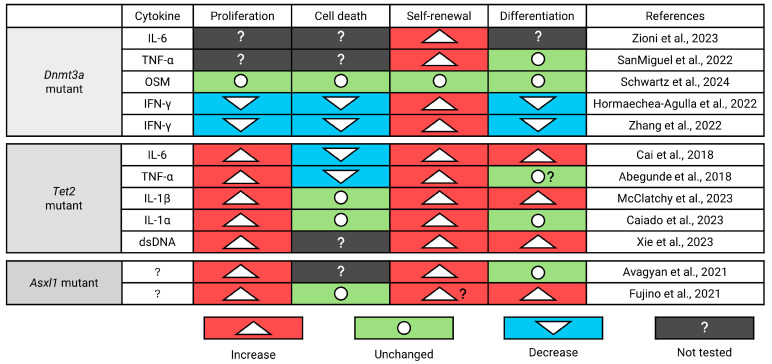
Summary of known differentially regulated cell fates of *Dnmt3a-*, *Tet2-*, and *Asxl1*-mutant mouse HSPCs in response to pro-inflammatory cytokines [71,90,91,92,96,97,98,99,100,101,102,103,111]. IL-6: Interleukin-6; TNF-α: Tumor Necrosis Factor-α; OSM: Oncostatin M; IFN-γ: Interferon-γ; IL-1β: Interleukin-1β; dsDNA: Double stranded DNA.

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
