# Peer review of "Decoding Clonal Hematopoiesis: Emerging Themes and Novel Mechanistic Insights"

_cancers, 2024, doi:10.3390/cancers16152634_

Round 1

Reviewer 1 Report

Comments and Suggestions for Authors

I enjoyed reading this comprehensive review by Shalmali Pendse and Dirk Loeffler providing a complete overview of clonal hematopoeisis especially in inflammatory environments. The review provides a well-written overview which is in my opinion useful to the field as a reference to direct future research for the challenging questions left. 

Author Response

Ref: cancers-3103780

Manuscript title: “Decoding clonal hematopoiesis: Emerging themes and novel  mechanistic insights

We appreciate the feedback and thank the Reviewers for their suggestions. We addressed all relevant comments of the reviewers as listed below.

Reviewer 1:

I enjoyed reading this comprehensive review by Shalmali Pendse and Dirk Loeffler providing a complete overview of clonal hematopoeisis especially in inflammatory environments. The review provides a well-written overview which is in my opinion useful to the field as a reference to direct future research for the challenging questions left. 

[Response to reviewer]: Thank you.

Reviewer 2 Report

Comments and Suggestions for Authors

The manuscript by Pendse and Loeffler summarizes the current literature on clonal hematopoiesis. The authors start with a very successful but somewhat long historical overview of the discovery of this phenomenon before they describe the current knowledge of the cells that trigger CH and the mechanism of clonal expansion. Finally, the influence of CH on the fate decision of stem cells and DNA methylation is discussed. 

The review is long, but overall very successful and easy to read. I also found the graphical presentation of the literature in thematically organized blocks very positive. Overall, this review is well written and entertaining to read. I only have a few minor comments on the manuscript that should be corrected:

1. In general, the abbreviations CH and CHIP are used interchangeably and at random. The authors should either use only one abbreviation or clearly define the abbreviations they use.

2. In line 319, "feedback look" should certainly be "feedback loop" 

3. In general, it is not clear from the text why proteins/genes are sometimes written in capital letters, i.e. referring to human proteins/genes, and sometimes only the first letter is capitalized. For example, "ATF3" in line 376 and "Atf3" in italics in line 379. Here, the authors should adhere to the nomenclature of protein and gene designations and also try to indicate the species in which the respective experiment was carried out.

4. In line 366, "chronic after M. avium infection model" should probably be "in a chronic model following M. avium infection".

5. In line 469, "proliferation amount" should certainly be "proliferation account".

6. In line 511, "self-renewal ability exhaust" should read "self-renewal ability and exhaust".

Comments on the Quality of English Language

The English language is very good. The only minor comments are noted in the report.

Author Response

Ref: cancers-3103780

Manuscript title: “Decoding clonal hematopoiesis: Emerging themes and novel  mechanistic insights

We appreciate the feedback and thank the Reviewers for their suggestions. We addressed all relevant comments of the reviewers as listed below.

Reviewer 2:

The manuscript by Pendse and Loeffler summarizes the current literature on clonal hematopoiesis. The authors start with a very successful but somewhat long historical overview of the discovery of this phenomenon before they describe the current knowledge of the cells that trigger CH and the mechanism of clonal expansion. Finally, the influence of CH on the fate decision of stem cells and DNA methylation is discussed. The review is long, but overall very successful and easy to read. I also found the graphical presentation of the literature in thematically organized blocks very positive. Overall, this review is well written and entertaining to read. I only have a few minor comments on the manuscript that should be corrected:

[Response to reviewer]: Thank you, we appreciate the feedback and a glad to see that the reviewer likes our manuscript.

  1. In general, the abbreviations CH and CHIP are used interchangeably and at random. The authors should either use only one abbreviation or clearly define the abbreviations they use.

[Response to reviewer]: Thank you, the reviewer’s comment is correct. We have corrected this and added the definition of CHIP in Lines 55 and 56: “CHIP was originally defined as a subset of CH with a variant allele frequency ≥2% of somatic mutations in genes associated with hematological malignancies.”

For the remainder of the text, we use the term “Clonal Hematopoiesis”.

  1. In line 319, "feedback look" should certainly be "feedback loop" 

[Response to reviewer]: Thank you for pointing this out. We have replaced “look” with “loop”.

  1. In general, it is not clear from the text why proteins/genes are sometimes written in capital letters, i.e. referring to human proteins/genes, and sometimes only the first letter is capitalized. For example, "ATF3" in line 376 and "Atf3" in italics in line 379. Here, the authors should adhere to the nomenclature of protein and gene designations and also try to indicate the species in which the respective experiment was carried out.

[Response to reviewer]: Thank you for bringing this to our attention. We went carefully through the entire text and corrected gene and protein names according to the official nomenclature. As suggested by the reviewer, we now also mention the species in several text passages to clarify whether we are talking about zebrafish, mouse, or human genes/proteins.

  1. In line 366, "chronic after M. avium infection model" should probably be "in a chronic model following M. avium infection".

[Response to reviewer]: Thank you for noticing this error: We have corrected this, the new sentence reads as follows: “However, in a mouse model of chronic infection using M. avium, IFN-γ signaling led to the downregulation of Socs3 and Nr4a1 in Dnmt3a-/- HSPCs”.

  1. In line 469, "proliferation amount" should certainly be "proliferation account".

[Response to reviewer]: We have replaced the word “amount” with “account”.

  1. In line 511, "self-renewal ability exhaust" should read "self-renewal ability and exhaust".

[Response to reviewer]: We have inserted the word “and” between the words “ability” and “exhaust”.

Reviewer 3 Report

Comments and Suggestions for Authors

In this current review “Decoding clonal hematopoiesis: Emerging themes and novel  mechanistic insightsThe review is well written and authors comprehensively explains different cellular mechanisms and proinflammatory factors affecting clonal evolution. The review sums up the topic very efficiently and can be published by incorporating of the following points

1) Authors should mention about notch signaling that is essential for the establishment of the earliest embryonic HSCs and its fate.

2) Authors should mention few lines related to the association of HSC with the innate immunity and relevance.

3)Figure 8 looks blurred can be more refined

4) There should be uniformity in the writing style of cytokines

5) Authors should briefly discuss that having knowledge of these mechanistic aspects can serve as a source for the treatment of certain diseases.

6) Authors should specifically mention about few of the diseases that can be cured attaining the knowledge from this compiled review.

Author Response

Ref: cancers-3103780

Manuscript title: “Decoding clonal hematopoiesis: Emerging themes and novel  mechanistic insights

We appreciate the feedback and thank the Reviewers for their suggestions. We addressed all relevant comments of the reviewers as listed below.

Reviewer 3:

In this current review “Decoding clonal hematopoiesis: Emerging themes and novel  mechanistic insightsThe review is well written and authors comprehensively explains different cellular mechanisms and proinflammatory factors affecting clonal evolution. The review sums up the topic very efficiently and can be published by incorporating of the following points

[Response to reviewer]: We are pleased to hear that our text was well received and appreciate the reviewer's time and effort to provide feedback.

1) Authors should mention about notch signaling that is essential for the establishment of the earliest embryonic HSCs and its fate.

[Response to reviewer]: Thank you for the suggestion. After carefully reading our manuscript again, we decided that highlighting notch signaling alone would be biased without including a broader and more detailed discussion of other factors. As a detailed discussion about factors regulating the earliest embryonic HSCs is beyond the scope of this review, we choose to avoid this bias and do not mention notch signaling specifically.   

2) Authors should mention few lines related to the association of HSC with the innate immunity and relevance.

[Response to reviewer]: Thank you. We already discuss the role of the innate immune system in the section “Mature mutant myeloid cells produce aberrant levels of inflammatory cytokines”. However, as suggested by the reviewer we extended this discussion and now also discuss how innate immune cells regulate HSC quality through trogocytosis

3) Figure 8 looks blurred can be more refined

[Response to reviewer]: We have uploaded the refined Figure 8.

4) There should be uniformity in the writing style of cytokines

[Response to reviewer]: Thank you for pointing this out. We have corrected these errors and all cytokines are now named in uniformity.

5) Authors should briefly discuss that having knowledge of these mechanistic aspects can serve as a source for the treatment of certain diseases.

[Response to reviewer]: Thank you for the suggestion. We now include a brief discussion, highlighting why mechanistic insights matter and that they can serve as a source for treatment (Line 551-558).

6) Authors should specifically mention about few of the diseases that can be cured attaining the knowledge from this compiled review.

[Response to reviewer]: Thank you. In the newly added paragraph mentioned in 5) we now mention some of these diseases.